# Reduced Sulfation Enhanced Oxytosis and Ferroptosis in Mouse Hippocampal HT22 Cells

**DOI:** 10.3390/biom10010092

**Published:** 2020-01-06

**Authors:** Haruna Nagase, Yasuhiro Katagiri, Kentaro Oh-hashi, Herbert M. Geller, Yoko Hirata

**Affiliations:** 1United Graduate School of Drug Discovery and Medical Information Sciences, Gifu University, Yanagido, Gifu 501-1193, Japan; haruna.science@gmail.com (H.N.); oohashi@gifu-u.ac.jp (K.O.-h.); 2Laboratory of Developmental Neurobiology, National Heart, Lung, and Blood Institute, NIH, Bethesda, MD 20892, USA; katagiry@nhlbi.nih.gov (Y.K.); gellerh@nhlbi.nih.gov (H.M.G.); 3Department of Chemistry and Biomolecular Science, Faculty of Engineering, Gifu University, Yanagido, Gifu 501-1193, Japan

**Keywords:** oxidative stress, oxytosis, ferroptosis, proteoglycans, sodium chlorate, sulfation

## Abstract

Sulfation is a common modification of extracellular glycans, tyrosine residues on proteins, and steroid hormones, and is important in a wide variety of signaling pathways. We investigated the role of sulfation on endogenous oxidative stress, such as glutamate-induced oxytosis and erastin-induced ferroptosis, using mouse hippocampal HT22 cells. Sodium chlorate competitively inhibits the formation of 3′-phosphoadenosine 5′-phosphosulfate, the high energy sulfate donor in cellular sulfation reactions. The treatment of HT22 cells with sodium chlorate decreased sulfation of heparan sulfate proteoglycans and chondroitin sulfate proteoglycans. Sodium chlorate and β-d-xyloside, which prevents proteoglycan glycosaminoglycan chain attachment, exacerbated both glutamate- and erastin-induced cell death, suggesting that extracellular matrix influenced oxytosis and ferroptosis. Moreover, sodium chlorate enhanced the generation of reactive oxygen species and influx of extracellular Ca^2+^ in the process of oxytosis and ferroptosis. Interestingly, sodium chlorate did not affect antioxidant glutathione levels. Western blot analysis revealed that sodium chlorate enhanced erastin-induced c-Jun N-terminal kinase phosphorylation, which is preferentially activated by cell stress-inducing signals. Collectively, our findings indicate that sulfation is an important modification for neuroprotection against oxytosis and ferroptosis in neuronal hippocampal cells.

## 1. Introduction

Alzheimer’s disease and Parkinson’s disease are the two most common neurodegenerative diseases to have become increasingly prevalent, especially in the grayer society. These diseases are characterized by a progressive neuronal loss in which many factors are involved in neuronal cell death, including oxidative stress. Cellular models of endogenous oxidative stress such as oxytosis and ferroptosis are an iron-dependent form of non-apoptotic cell death, and reproduce many of the abnormal characteristics in these neurodegenerative diseases [1]. It has been found that excessive extracellular glutamate or erastine inhibits the cystine/glutamate antiporter and depletes intracellular cysteine, leading to a reduction in glutathione (GSH) content and the accumulation of reactive oxygen species (ROS), which eventually cause oxidative stress-induced cell death or oxytosis/ferroptosis in HT22 cells [2,3,4,5]. HT22 is an immortalized mouse hippocampal cell line which is widely used as in vitro model for studying endogenous oxidative stress. Glutamate induces not only oxidative stress-induced cell death but also excitotoxic cell death through ionotropic glutamate receptors in neurons. Because HT22 cells lack functional ionotropic glutamate receptors, excitotoxicity can be excluded as a cause for glutamate-triggered cell death [2].

Previous reports have demonstrated that various natural and synthetic compounds against ROS-mediated damage prevented glutamate-induced endogenous oxidative stress [6,7,8]. These mainly target generation and accumulation of ROS within cells. On the other hand, many findings evidenced antioxidant properties of sulfated glycosaminoglycans (GAGs), particularly for heparan sulfate (HS) and chondroitin sulfate (CS) [9,10], which may reduce oxidative damage extracellularly [11,12]. The majority of GAGs linked to protein to form proteoglycans such as heparan sulfate-proteoglycans (HSPGs) and chondroitin sulfate-proteoglycans (CSPGs) are located primarily on the cell surface or in the extracellular matrix (ECM) that lead to highly negatively charged structures. This cellular microenvironment might potentially be able to scavenge and bind cations, which may provide some neuroprotection against oxidative stress [13].

In this paper, we aimed to evaluate the effect of reduced sulfation on endogenous oxidative stress-induced cell death in mouse hippocampal HT22 cells using sodium chlorate, a widely used inhibitor of sulfation [14,15], and β-d-xyloside, which prevents the attachment of sulfated GAG chains to core proteins [16]. The treatment of cells with sodium chlorate in order to reduce the sulfation of proteoglycans is suggested to be a useful method for examining possible functions of the sulfate groups [14,15,17], while changes with xylosides implicate GAG chains in a process. We found that both treatments enhanced oxidative stress.

## 2. Materials and Methods

### 2.1. Materials

Erastin (Sigma-Aldrich Corporation, St. Louis, MO, USA, Cat# E7781) and 4-methylumbelliferyl-β-d-xylopyranoside (Sigma-Aldrich Corporation, St. Louis, MO, USA, Cat# M7008) were dissolved in cell culture-grade dimethyl sulfoxide and stored in the dark at −20 °C. Phthaldialdehyde (Sigma-Aldrich Corporation, St. Louis, MO, USA, Cat# P0657) was dissolved in methanol and freshly prepared. Sodium chlorate (FUJIFILIM Wako Pure Chemical, Osaka, Japan, Cat# 193-01642) was dissolved in Dulbecco’s modified Eagle’s medium (DMEM; FUJIFILM Wako Pure Chemicals, Osaka, Japan, Cat# 041-29775) and freshly prepared.

### 2.2. Cell Culture

HT22 immortalized hippocampal cells (RRID:CVCL_0321) were kindly provided by Dr. Schubert (The Salk Institute, La Jolla, CA, USA). The HT22 cells were cultured in DMEM (FUJIFILM Wako Pure Chemicals, Osaka, Japan) supplemented with 5% fetal bovine serum (HyClone Laboratories, Logan, UT, USA, Cat# SH30406.02) at 37 °C in a 5% CO_2_ incubator.

### 2.3. Immunofluorescent Staining and Imaging

To evaluate HS and CS GAG chains, HT22 cells were stained with either monoclonal anti-HS antibody 10E4 (F58-10E4 clone, 1:250 dilution; AMS Biotechnology Ltd. Abingdon, UK, RRID:AB_10891554) or monoclonal anti-CS antibody CS-56 (anti-6S, 2S, 4S antibody) (1:100 dilution; Sigma-Aldrich Corporation, St. Louis, MO, USA, RRID:AB_476879). HT22 cells (2.0 × 10^3^/cm^2^) were grown on 24-well plate and treated with sodium chlorate. The cells were fixed with 4% paraformaldehyde for 15 min and then non-specific binding was blocked with 2.5% bovine serum albumin (BSA) in phosphate-buffered saline (PBS). Subsequently, HT22 cells were incubated in primary antibodies with 1% BSA or immunoenhancing reagent (Can Get Signal^®^ Immunostain Solution B; TOYOBO, Osaka, Japan, Cat# NKB-401) overnight at 4 °C. HT22 cells were washed with PBS and immersed in 2.5% BSA or Can Get Signal^®^ Immunostain Solution B (TOYOBO, Osaka, Japan) containing anti-IgM-FITC (fluorescein isothiocyanate, 1:250 dilution; Jackson ImmunoResearch, PA, USA) and then incubated with 10 µg/mL Hoechst 33258 (Molecular Probes, OR, USA) for nuclei staining. After washing with PBS, fluorescence images were captured with a fluorescence digital microscope (BZ-X800, Keyence, Osaka, Japan). The fluorescent intensity was quantified using Keyence image measurement and the analyzing software (BZ-X800 Analyzer, Keyence Corporation, Osaka, Japan).

### 2.4. Western Blotting

HT22 cells were lysed with sodium dodecyl sulfate (SDS) sample buffer (62.5 mM Tris-HCl (pH 6.8), 2% SDS, 10% glycerol, 0.1% bromophenol blue). Protein concentration in lysate was determined by the DC Protein Assay Kit (Bio-Rad Laboratories, Hercules, CA, USA, Cat# 5000111JA) using γ-globulins as a standard. Equal amounts of protein (20 µg) were separated by SDS-PAGE under reducing conditions and transferred to nitrocellulose membranes. Membranes were blocked with 5% skim milk in PBS containing 0.05% Tween-20 (PBS-T) for 1 h at room temperature. Membranes were incubated with the following primary antibodies at the indicated dilutions: anti-phospho-SAPK/JNK (Thr183/Tyr185) (81E11) (1:1000 dilution, rabbit monoclonal, Cell Signaling Technology, Inc., Beverly, MA, USA, RRID:AB_823588), anti-HO-1 (1:2000 dilution, mouse monoclonal, Enzo Life Science, Inc., Farmingdale, NY, USA, RRID:AB_10617276), and anti-glyceraldehyde 3 phosphate dehydrogenase (GAPDH; 1:5000 dilution; mouse monoclonal, Acris Antibodies, Inc., San Diego, CA, USA, RRID:AB_1616730). Membranes were washed and incubated with an anti-mouse or anti-rabbit IgG secondary antibody conjugated to horseradish peroxidase (1:2000 dilution, Cell Signaling Technology, Inc.) in blocking buffer. Signals were visualized using Amersham ECL Western Blotting Detection Reagent (GE Healthcare UK Ltd., Buckinghamshire, England, Cat# RPN2106) or SuperSignal^TM^ West Dura Extended Substrate (Thermo Fisher Scientific, Waltham, MA, USA, Cat# 34075) and Amersham Hyperfilm ECL (GE Healthcare UK Ltd., Buckinghamshire, UK, Cat# 28906837). The blots were reprobed with different antibodies after stripping in 62.5 mM Tris-HCl (pH 6.7), 100 mM β-mercaptoethanol, and 2% SDS at 50 °C for 30 min, when required. Quantification of the bands was performed using ImageJ.

### 2.5. Cell Viability Assay

Cell viability was determined with 2-(4-Iodophenyl)-3-(4-nitrophenyl)-5-(2,4-disulfophenyl)-2H-tetrazolium (WST-1) using a Cell counting Kit (Dojindo Laboratories, Kumamoto, Japan, Cat# CK01). HT22 cells were cultured on 96-well plates and treated with sodium chlorate for the indicated period. Cultured medium was replaced to the medium containing 500 nM WST-1 and incubated for additional 2 h at 37 °C in a 5% CO_2_ incubator. Absorbance was measured at 450 nm with 690 nm as a reference wavelength.

### 2.6. Cell Death Assay

Cytotoxicity was determined by measuring lactate dehydrogenase (LDH) released into the culture medium with a Cytotoxicity Detection Kit according to the manufacture’s protocol (Takara Bio Inc., Shiga, Japan, Cat# MK401). HT22 cells were grown on 48-well plates in 400 µL of the medium. After the treatment of the cells under various experimental conditions, 10 µL of the culture medium was analyzed for LDH activity. The percent cell death was calculated as 100 × (experimental release − blank)/(total release − blank), where total release is the activity from 10 µL of cells lysed with 1% Triton X-100.

### 2.7. GSH Determination

HT22 cells were cultured on 6-well plates and treated under various experimental conditions. The cells were collected with cold PBS and centrifuged. The cell pellet was resuspended in 120 μL of 0.1 M sodium phosphate buffer (pH 8.0)/5 mM EDTA, and deproteinized by adding four volumes of 25% (*w*/*v*) metaphosphoric acid. The sample was centrifuged for 10 min at 15,000× *g*. The supernatant was used for GSH assay developed by Hissin and Hilf [18] with slight modification. Briefly, the supernatant (5 µL), 185 µL of 0.1 M sodium phosphate buffer (pH 8.0) containing 5 mM EDTA and 10 µL of phthaldialdehyde solution (1 mg/mL in methanol) were added to a 96-well black plate and incubated at 25 °C for 15 min. Fluorescence intensity was measured at 420 nm with excitation at 350 nm using a microplate reader (Varioskan Flash; Tharmo Fischer Scientific, Waltham, MA, USA). The resultant pellet was solubilized in 75 µL 0.2 M NaOH and used for protein assay. GSH was normalized to cellular protein measured by the DC Protein Assay (Bio-Rad Laboratories, Hercules, CA, USA).

### 2.8. ROS Production

HT22 cells were cultured on a 12-well plate and treated under various experimental conditions. MitoSOX (5 µM; Thermo Fisher Scientific, Waltham, MA, USA, Cat# M36008) was added and incubated for 15 min at 37 °C. The medium was replaced by serum-free DMEM without phenol red (Thermo Fisher Scientific, Waltham, MA, USA, Cat# 21063029). The fluorescence was measured by a fluorescence digital microscope (BZ-X800, Keyence Corporation, Osaka, Japan). Fluorescent intensity was quantified using Keyence image measurement and the analyzing software (BZ-X800 Analyzer, Keyence Corporation, Osaka, Japan).

### 2.9. Ca^2+^ Influx

HT22 cells were grown on a 12-well plate and treated under various experimental conditions. Fluo4-AM (2 μM; Thermo Fisher Scienctific, Waltham, MA, USA, Cat# F14201) and Pluronic F-127 (0.04%; Thermo Fisher Scientific, Waltham, MA, USA, Cat# P3000MP) were added and incubated for 15 min at 37 °C. The medium was replaced by serum-free DMEM without phenol red. The fluorescence was measured by a fluorescence digital microscope (BZ-X800, Keyence Corporation). The fluorescent intensity was quantified using Keyence image measurement and the analyzing software (BZ-X800 Analyzer, Keyence Corporation, Osaka, Japan).

### 2.10. Statistical Analyses

The numerical data were statistically analyzed with GraphPad Prism 6.0 (GraphPad Software, Inc., La Jolla, CA, USA, PRID:SCR_002798). The significant differences between the control group and experimental group were determined by Bonferroni’s multiple comparison tests following ANOVA.

## 3. Results

### 3.1. Effect of Sodium Chlorate on Cell Viability in HT22 Cells

We first examined the cytotoxicity of sodium chlorate in HT22 cell (Figure 1). Cell viability was not changed up to 40 mM and decreased significantly at 60 mM after the incubation of the cells for 24 h and 48 h. Accordingly, we chose non-toxic (20 mM) and slightly toxic (60 mM) concentrations of sodium chlorate for further experiments.

### 3.2. Sodium Chlorate Reduced the Sulfation of GAGs

Proteoglycan GAG chains are highly sulfated. To check the efficacy of sodium chlorate treatment, we investigated the effect of sodium chlorate on the sulfation of HS and CS GAG chains in HT22 cells (Figure 2). HS was detected by the 10E4 antibody (F58-10E4 clone) which recognize common epitopes on HS including an N-sulfated glucosamine residue [19]. CS was detected by the CS-56 antibody which is specific for the GAG portion of native CSPGs. Treatment with 20 mM sodium chlorate decreased both HS and CS, suggesting that sodium chlorate reduced sulfation of HSPGs and CSPGs (Figure 2a,b).

### 3.3. Sodium Chlorate Treatment Enhanced Extracellular Glutamate- and Erastin-Induced Cell Death in HT22 Cells

Next, we examined the effect of sodium chlorate on glutamate- and erastin-induced oxidative stress. Because 10 mM glutamate and 0.5–1 µM erastin caused nearly maximal cell death (data not shown), we chose 5 mM and 0.2 µM, respectively, as submaximal concentrations. Sodium chlorate enhanced both glutamate- and erastin-induced cell death in a concentration-dependent manner (Figure 3). Although cell death by LDH assay in Figure 3 showed no increase up to 60 mM sodium chlorate, total LDH activity which is proportional to cell number decreased (data not shown), indicating that cell proliferation was suppressed under this condition. The data are consistent with the result obtained from cell viability by WST-1 assay shown in Figure 1. These results suggest that reduced sulfation exacerbated both glutamate-induced oxytosis and erastin-induced ferroptosis. Because erastin-induced cell death was more affected by sodium chlorate than glutamate-induced cell death, we focused on erastin-induced cell death, ferroptosis, for further experiments.

### 3.4. β-d-Xyloside Enhanced Extracellular Glutamate- and Erastin-Induced Cell Death in HT22 Cells

Although sodium chlorate reduced HS and CS efficiently in HT22 cells, it potentially inhibits sulfation of not only extracellular glycans but also tyrosine residues on proteins and steroid hormones, which is important in many signaling pathways. Therefore, we used 4-methylumbelliferyl-β-d-xylopyranoside (β-d-xyloside), a compound which interferes with proteoglycan synthesis by acting as an artificial acceptor for glycosaminoglycan synthesis and thereby competing with the proteoglycan core protein [20]. The treatment of the cells with β-d-xyloside also caused a reduction in HS and CS immunoreactivity as well as exacerbation of erastin-induced ferroptoosis (Figure 4a,b). These findings confirmed that the reduced synthesis of HS and CS enhanced endogenous oxidative stress-induced cell death.

### 3.5. Sodium Chlorate Treatment Enhanced Erastin-Induced ROS Production but Did Not Affect GSH Depletion in HT22 Cells

Earlier studies have demonstrated that the glutamate- and erastin-induced oxidative stress is initiated by inhibiting the transmembrane cystine/glutamate antiporter, leading to depletion of intracellular GSH, accumulation of ROS, and excess influx of Ca^2+^, resulting in cell death in HT22 cells [2,3,4,5]. To investigate whether reduced sulfation also affects to the cell death cascade, intracellular GSH, accumulation of ROS and an influx of Ca^2+^ were examined after the treatment with erastin in the presence or absence of sodium chlorate. As reported previously, erastin increased the production of ROS which was enhanced by treatment with sodium chlorate (Figure 5b). Similarly, an erastin-induced influx of Ca^2+^ was accelerated by sodium chlorate (Figure 5c). These results suggested that reduced sulfation is accompanied by increased ROS production and Ca^2+^ influx, leading to the exacerbation of ferroptotic cell death. In contrast, GSH levels decreased similarly in erastin-treated HT22 cells in the presence or absence of sodium chlorate (Figure 5a). These results indicate that reduced sulfation did not affect cystine-glutamate transport via system xc-, a cystine/glutamate antiporter.

### 3.6. Effect of ROS Scavenging and Iron Chelating Chemicals on Sodium Chlorate-Enhanced Ferroptotic Cell Death

Several studies have reported that ROS scavenging and iron chelating chemicals such as α-tocopherol and deferoxamine prevent glutamate- or erastin-induced cell death in HT22 cells [21,22,23]. We examined whether ROS scavenging and iron chelating chemicals mitigated enhanced toxicity of sodium chlorate. Cotreatment with α-tocopherol (1 µM) completely suppressed erastin-induced cell death irrespective of the presence or absence of sodium chlorate. In contrast, the protective effect of α-tocopherol (0.1 µM or less) weakened significantly in the presence of sodium chlorate (Figure 6a). Similarly, deferoxamine (5 µM or more) completely suppressed erastin-induced cell death, however, deferoxamine (2 µM) only partially prevented enhanced toxicity of sodium chlorate (Figure 6b). These results indicate that increased ROS and Fe^2+^ are involved in sodium chlorate-enhanced ferroptotic cell death.

### 3.7. Sodium Chlorate Treatment Enhanced Erastin-Induced Phosphorylation of c-Jun N-Terminal Kinase (JNK) in HT22 Cells

Oxidative stress directly or indirectly contributes to ROS-mediated various intracellular signaling pathway. ROS activates the Nrf2-antioxidant response element (ARE) pathway, an indicator and modulator of oxidative stress in neurodegeneration [24] that induces the expression of downstream target genes, such as hemoxygenase-1 (HO-1). ROS also induces phosphorylation of the stress kinase JNK which is involved in cellular responses to environmental stresses. We investigated whether sodium chlorate affected these intracellular signaling pathways in response to oxidative stress. Consistent with the previous reports [25], phosphorylation of JNK was increased upon the treatment with erastin and this induced phosphorylation was further enhanced by sodium chlorate (Figure 7a–c). In contrast, the erastin-induced increase in HO-1 expression was not changed significantly by treatment with sodium chlorate (Figure 7a,d). These results suggest that the activation of the JNK pathway is involved in reduced-sulfation enhanced oxidative stress, whereas the Nrf2-ARE pathway is not influenced by reduced sulfation in the cells.

## 4. Discussion

In this study, we examined the role of sulfation in oxidative stress using mouse hippocampal HT22 cells, a well-established cellular model for studying endogenous oxidative stress [4]. We found that reduced sulfation exacerbated both glutamate-induced oxytosis and erastin-induced ferroptosis via enhancing ROS production and Ca^2+^ influx. Our results suggest that the reduced sulfation influences the progression of oxidative stress-induced cell death, which occurs in a variety of neurodegenerative diseases.

Sodium chlorate appeared to be the most effective of the substance used in reducing the sulfation of a variety of macromolecules [26]. On the other hand, sodium chlorate, a herbicide and major water disinfectant byproduct, generates reactive oxygen species and induces oxidative damage in human erythrocytes [27]. Treatment of HT22 cells with sodium chlorate alone did not induce ROS production (Figure 5b) and did not affect cellular antioxidant system such as GSH content (Figure 5a). Ali et al. proposed that chlorate enters the erythrocyte, probably through band 3 anion channel, where it is reduced to chloride ion and this process generates ROS and reactive nitrogen species [27]. However, at present, it is uncertain whether band 3 anion channel can function in HT22 cells. Because chlorate is an in vitro inhibitor of ATP-sulfurylase, the first enzyme in the biosynthesis of 3′-phosphoadenosine 5′-phosphosulfate which is the ubiquitous co-substrate for sulfation [28], the exact mediators of ROS generation are not clear.

Previous studies have suggested that GAGs play an important role in the pathogenesis and alleviation of neurodegenerative disorders, including Alzheimer’s disease and Parkinson’s disease. In vitro studies showed that CSPGs protected excitotoxic cell death induced by glutamate in hippocampal and cortical primary neurons and suggested that CSPGs exert their neuroprotective action by antagonizing cellular responses following the activation of excitatory amino acid receptors rather than blocking the activation of certain receptor subtypes [29,30]. Whether this neuroprotection is dependent upon sulfation of the GAG chains is not clear. In this study, use of β-d-xyloside, an inhibitor of GAG chain synthesis [20], proved that reduced GAG chains on HSPGs and CSPGs influenced endogenous oxidative stress-induced cell death similarly to sodium chlorate. Other studies showed that HS had a significant neuroprotective effect in the face of ibotenic acid brain injury [31]. The buffering features of CSPGs are supported by their role on determining the local diffusion properties of calcium in the brain extracellular space [32]. This mechanism is also suggested to be involved in neuroprotection against iron-induced cell death by perineuronal nets, a specialized form of ECM which mainly consist of CSPGs [13]. Collectively, negative charges on HSPGs and CSPGs may influence the diffusion of divalent cations such as Ca^2+^ and Fe^2+^ which play a critical role in glutamate excitotoxicity. Here, we demonstrated that sodium chlorate weakened the protective effect of an iron chelator, deferoxamine, on erastin-induced cell death. Both Ca^2+^ and Fe^2+^ are involved intimately in glutamate- and erastin-induced cell death in HT22 cells [33], suggesting that reduced sulfation by the treatment with sodium chlorate enhances diffusion of divalent cations and exacerbates glutamate- and erastin-induced oxidative stress.

Western blot analysis revealed that sodium chlorate enhanced erastin-induced phosphorylation of JNK. Because it has been well established that ROS are potent inducers of JNK, it is probable that sodium chlorate enhanced JNK phosphorylation by increasing ROS production [34]. In addition, reduced sulfation may affect JNK phosphorylation extracellularly. It has been reported that the subendothelial ECM modulates JNK activation by flow [35]. These findings suggest that JNK activity could be modulated extracellularly by not only fluid shear stress, but also reduced sulfation. Taken together, our results support a role for sulfation in the maintenance of cell survival and prevention of oxidative-stress induced cell death.

## 5. Conclusions

We investigated the role of sulfation on endogenous oxidative stress such as glutamate-induced oxytosis and erastin-induced ferroptosis using mouse hippocampal HT22 cells. The widely used sulfation inhibitor sodium chlorate and β-d-xyloside, which prevents proteoglycan glycosaminoglycan chain attachment, both reduced HS and CS, and exacerbated glutamate- and erastin-induced cell death, suggesting that extracellular matrix influenced oxytosis and ferroptosis. Moreover, sodium chlorate enhanced the generation of ROS and influx of extracellular Ca^2+^ in the process of oxytosis and ferroptosis. Western blot analysis revealed that sodium chlorate enhanced erastin-induced JNK phosphorylation, which is preferentially activated cell stress-inducing signals. Our findings indicate that sulfation is an important modification for neuroprotection against oxytosis and ferroptosis in neuronal hippocampal cells.

## Figures and Tables

**Figure 1 biomolecules-10-00092-f001:**
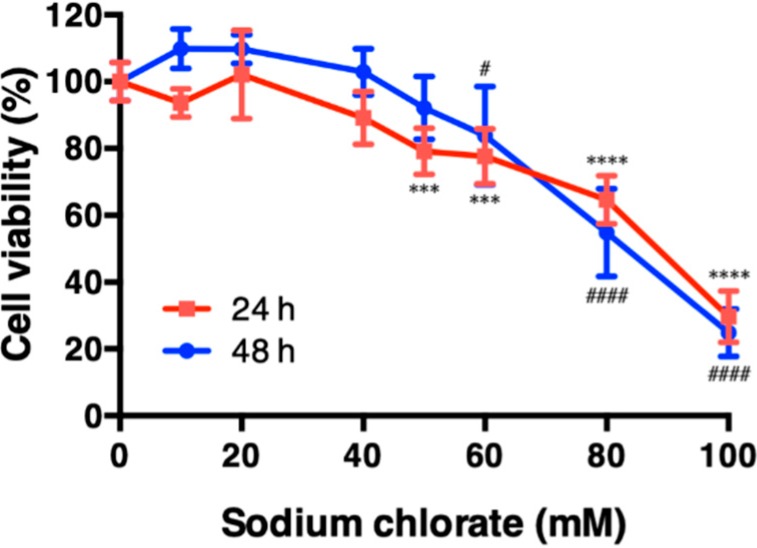
Effect of sodium chlorate on cell viability. HT22 cells were treated with the indicated concentrations of sodium chlorate for 24 h (red line) or 48 h (blue line) and cell viability was measured by WST-1 assay. The data are presented as the means ± SD of at least six independent cultures. *** *p* < 0.001, **** *p* < 0.0001 compared with the 24 h control; ^#^
*p* < 0.05, ^####^
*p* < 0.0001 compared with the 48-h control.

**Figure 2 biomolecules-10-00092-f002:**
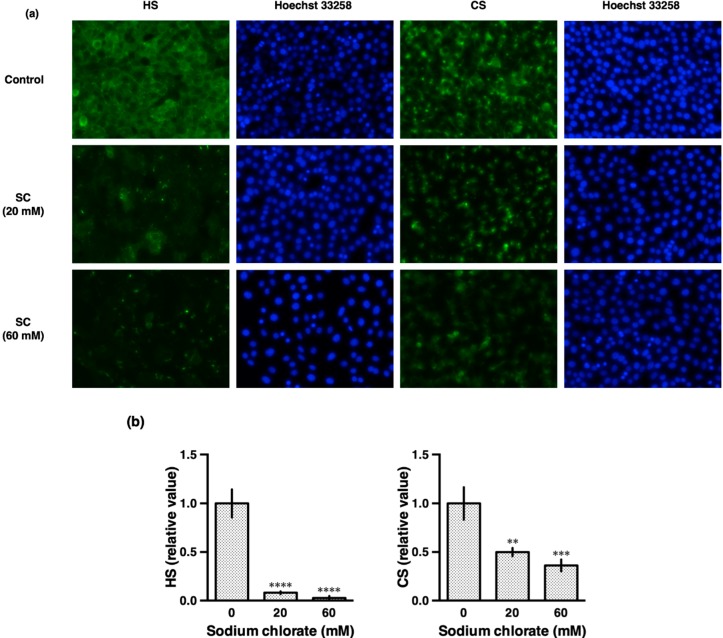
Effect of sodium chlorate on HS and CS in HT22 cells. (**a**) Representative fluorescence micrographs of typical data shown. HT22 cells were cultured with or without sodium chlorate (20 or 60 mM) for 24 h and subjected to immunofluorescent staining. HT22 cells were stained with 10E4 (anti-HS antibody) and CS-56 (anti-CS antibody). (**b**) Fluorescence intensity was quantified using a Keyence image measurement and analyzing software (VH-H1A5; Keyence). The data are presented as the mean ± SD (*n* = 4). ** *p* < 0.01, *** *p* < 0.001, **** *p* < 0.0001 compared with the control.

**Figure 3 biomolecules-10-00092-f003:**
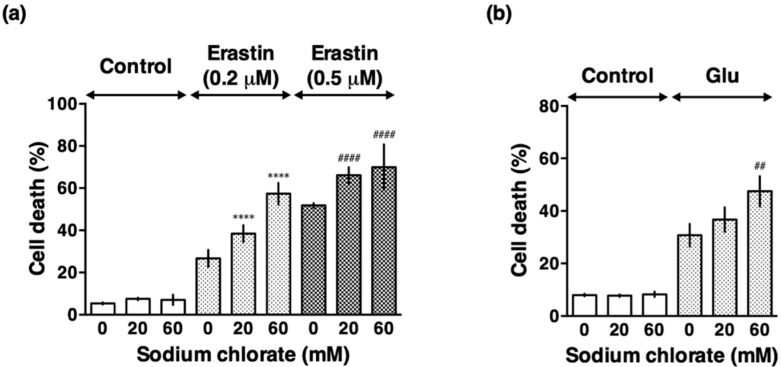
Sodium chlorate enhanced endogenous oxidative stress-induced cell death in HT22 cells. (**a**) Effect of sodium chlorate on erastin-induced cell death. HT22 cells were treated with 0.2 µM or 0.5 µM erastin in the presence or absence of sodium chlorate (20 or 60 mM) for 24 h and cell death was determined by LDH assay. (**b**) Effect of sodium chlorate on glutamate-induced cell death. HT22 cells were treated with 5 mM glutamate in the presence or absence of sodium chlorate (20 or 60 mM) for 24 h and cell death was determined by LDH assay. The data are presented as the mean ± SD. The data were obtained from at least four independent cultures. **** *p* < 0.0001, ^####^
*p* < 0.0001 compared with 0.2 µM or 0.5 µM erastin alone; ^##^
*p* < 0.01 compared with glutamate alone.

**Figure 4 biomolecules-10-00092-f004:**
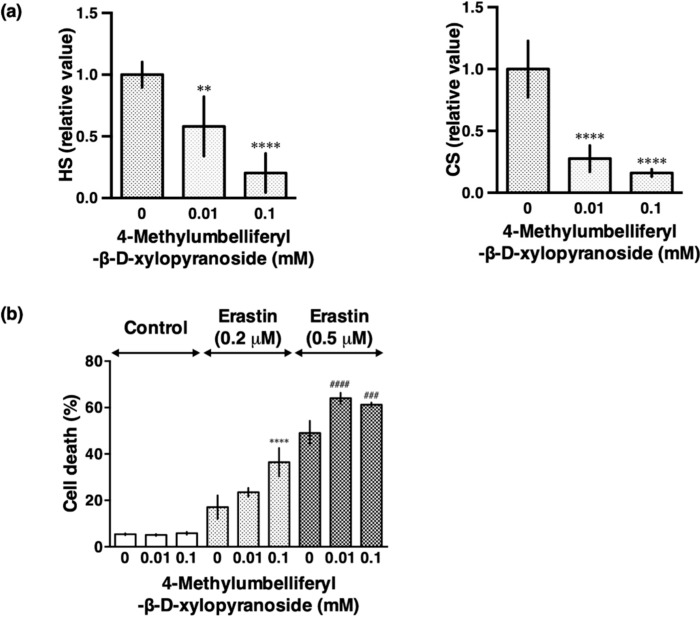
Effect of β-d-xyloside on HS and CS levels and erastin-induced cell death. To examine the effect of β-d-xyloside, HT22 cells were cultured with the indicated concentrations of 4-methylumbelliferyl-β-d-xylopyranoside for 2 days. The cells were plated on 24-well plates for immunofluorescent staining of HS and CS or 48 well plates for cell death assay. (**a**) HT22 cells were cultured for another 24 h in the presence of 4-methylumbelliferyl-β-d-xylopyranoside and stained with 10E4 (anti-HS antibody) and CS-56 (anti-CS antibody), respectively. Fluorescence intensity was quantified using a Keyence image measurement and analyzing software (VH-H1A5; Keyence). The data are presented as the mean ± SD (*n* = 7–9). ** *p* < 0.01, **** *p* < 0.0001 compared with the control. (**b**) HT22 cells were treated with 0.2 µM or 0.5 µM erastin in the presence or absence of 4-methylumbelliferyl-β-d-xylopyranoside (0.01 or 0.1 mM) for 24 h and cell death was determined by LDH assay. The data are presented as the mean ± SD. The data were obtained from at least four independent cultures. **** *p* < 0.0001 compared with 0.2 µM erastin alone; ^###^
*p* < 0.001, ^####^
*p* < 0.0001 compared with 0.5 µM erastin alone.

**Figure 5 biomolecules-10-00092-f005:**
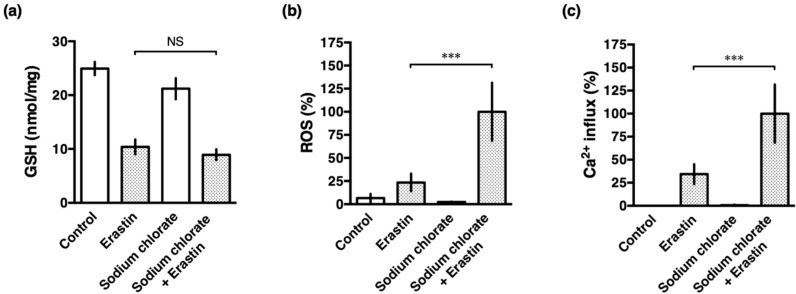
Effects of sodium chlorate on GSH contents, ROS production, and Ca^2+^ influx in erastin-treated HT22 cells. The cells were treated with 0.2 µM erastin in the presence or absence of 20 mM sodium chlorate for 8 h. (**a**) Glutathione contents were measured and normalized to the protein concentration. (**b**) ROS production was determined with MitoSOX reagent. (**c**) Intracellular Ca^2+^ was detected with Fluo-4 reagent. The data are presented as the mean ± SD. The data were obtained from at least three independent cultures. *** *p* < 0.001; NS, not significant.

**Figure 6 biomolecules-10-00092-f006:**
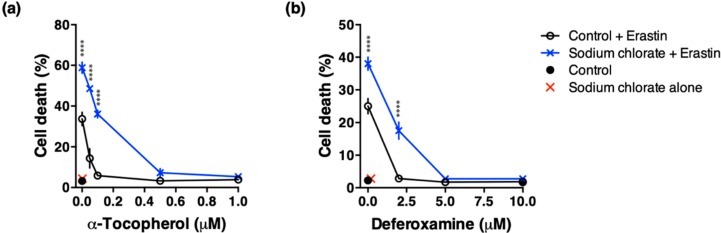
Effect of α-tocopherol or deferoxamine on erastin-induced cell death in the presence or absence of sodium chlorate. (**a**) ROS scavenging by α-tocopherol mitigated enhanced toxicity of sodium chlorate. (**b**) Iron chelating by deferoxamine mitigated enhanced toxicity of sodium chlorate. HT22 cells were treated with 0.2 µM erastin and indicated concentrations of α-tocopherol or deferoxamine in the presence or absence of 60 mM sodium chlorate for 24 h and cell death was determined by LDH assay. The data are presented as the mean ± SD. The data were obtained from at least four independent cultures. **** *p* < 0.0001 compared with control + erastin at the same concentration.

**Figure 7 biomolecules-10-00092-f007:**
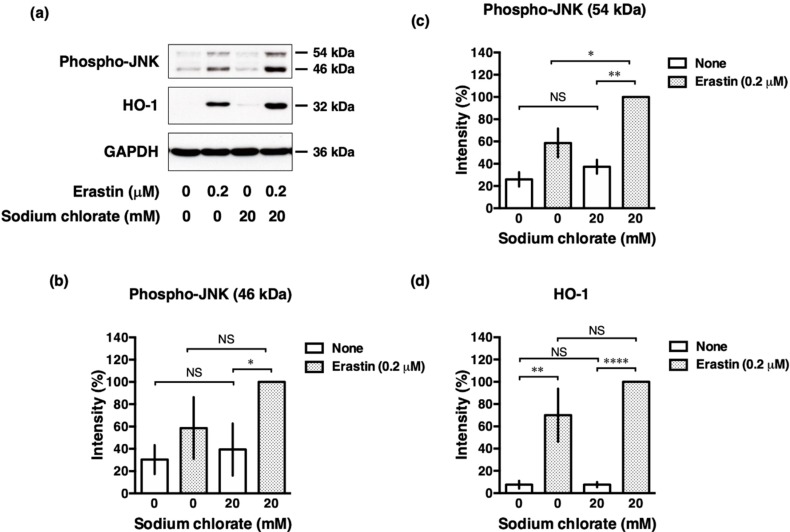
Reduced sulfation enhanced phosphorylation of JNK. (**a**) Western blot analysis of phospho-JNK and HO-1. (**b**–**d**) The band intensity was quantified using an Image J software. The data are presented as the mean ± SD. The data were obtained from at least three independent experiments. * *p* < 0.05, ** *p* < 0.01, **** *p* < 0.0001; NS, not significant.

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
