# Peer review of "Reduced Sulfation Enhanced Oxytosis and Ferroptosis in Mouse Hippocampal HT22 Cells"

_biomolecules, 2020, doi:10.3390/biom10010092_

Round 1

Reviewer 1 Report

The authors show that sodium chlorate (SC) decreases sulfation of heparan sulfate proteoglycans and chondroitin sulfate proteoglycans and that SC and beta-D24 xyloside, an inhibitor of proteoglycan glycosaminoglycan chain attachment, exacerbate glutamate- and erastin-induced cell death, suggesting that extracellular matrix influences oxytosis and ferroptosis, in mouse hippocampal HT22 cells. The authors also show that SC favors the generation of reactive oxygen species and influx of extracellular Ca2+, yet without affecting glutathione levels, and enhances erastin-induced JNK activation.

Comments

4. (c) should be (b). 6. Total JNK and p-JNK/total JNK rations should be shown instead of p-JNK/GADPH ratios. 6D. Western blots of HO-1 should be shown. A number of control experiments are missing. To support their conclusions, the authors should show that (1) ROS-scavenging chemicals reduce effects of SC and beta-D24 xyloside, and (2) pharmacological inhibition of JNK protects the cells against the detrimental effects of SC.

Author Response

Reviewer 1:

4. (c) should be (b).

 Corrected.

6. Total JNK and pJNK/total JNK rations should be shown instead of pJNK/GADPH ratios.

 We showed GAPDH simply as a loading control. The intensity of phosphor-JNK was measured as described in the method and in our previous paper (Hirata et al., Toxicology 418, 1-10, 2019).

6D. Western blots of HO1 should be shown.

 Western blots of HO-1 was shown in Fig. 7a.

A number of control experiments are missing. To support their conclusions, the authors should show that (1) ROS scavenging chemicals reduce effects of SC and beta D-xyloside, and (2) pharmacological inhibition of JNK protects the cells against the detrimental effects of SC.

 (1) We carried out additional experiments using a ROS scavenging chemical, α-tocopherol, and added the section 3.6. Effect of ROS scavenging and iron chelating chemicals on sodium chlorate-enhanced ferroptotic cell death and the data as Figure 6a.

 (2) We have reported previously that a pharmacological inhibitor of JNK SP600125 does not have a prominent protective effect on glutamate- and erastin-induced cell death (Hirata et al., Toxicology 418, 1-10, 2019, 2019). This is explained by the fact that binding of activated JNK to non-transcriptional targets such as the outer mitochondrial membrane protein Sab leads to increased mitochondrial ROS and mediates ER stress-induced apoptosis in cultured cells and ischemic necrosis in the brain (Chambers et al., J. Biol. Chem. 288, 1079-1087, 2013; Win et al., Cell Death Dis. 5, e989, 2014). Therefore non-transcriptional targets of phospho-JNK could play a significant role not only in ER stress but also in oxytosis and ferroptosis.

Reviewer 2 Report

Summary

This study investigated a neuroprotective role of sulfation on glycoproteins in the extracellular matrix (ECM) in HT22 cells.  Sodium chlorate and beta-D-xyloside were used to inhibit sulfation in ECM and its impact on glutamate- or erastin-induced toxicity was assessed.  Reduced sulfated glycosaminoglycans significantly enhanced glutamate- and erastin-induced cell death, erastin-induced ROS generation and Ca2+ influx.  Elevated ROS appeared to potentiate JNK pathway, while Nrf2 signaling was not enhanced by sodium chlorate.  Collectively, findings presented in this manuscript support an anti-oxidative role of sulfation in ECM (or glycosaminoglycans) in HT22 cells.

Major concerns

1. While cell viability by WST-1 assay (Fig. 1) indicates a significant, about 20% loss at 60 mM sodium chlorate for 24 hrs, cell death by LDH assay in Fig. 2 shows no increase up to 60 mM. Although WST-1 assay and LDH assay are measuring different factors (mitochondrial activity vs. LDH released extracellular space), the authors may need to explain this potential discrepancy.

2. Please discuss why sodium chlorate (or reduction of sulfation in ECM) treatment did not increase background ROS generation (e.g. Fig. 5b)

3. What is the interpretation of increased phospho-p54-JNK but not phospho-p46-JNK following co-treatment with erastin and sodium chlorate over erastin alone (Fig. 6)? Will inhibiting JNK but not Nrf2 ameliorate erastin + sodium chlorate toxicity and cell death in HT22 cells?

4. If diffusion of divalent cation (e.g. Fe2+) by HSPGs and CSPGs may play a critical role in the toxicity described in this study, will co-treatment with Fe chelator mitigate enhanced toxicity by sodium chlorate?

Minor concerns

1. Please label panel (b) in Figure 4.

2. Please define system Xc- as cysteine/glutamate antiporter on first appearance.

3. Line 299-304 where the authors are explaining the mechanism of action of sodium chlorate is a bit confusing. Please edit this.

4. Line 310-313 is a repeated information found in the introduction, and citations found around (e.g. #2, 20, and 14) are using ( ) instead of [ ].

Author Response

Reviewer 2:

Major concerns

1. While cell viability by WST1 assay (Figure 1) indicates a significant, about 20% loss at 60 mM sodium chlorate for 24 hrs, cell death by LDH assay in Fig. 2 shows no increase up to 60 mM. Although WST1 assay and LDH assay are measuring different factors (mitochondrial activity vs. LDH released extracellular space), the authors may need to explain this potential discrepancy.

As pointed out by the reviewer, cell viability determined by WST-1 assay and cell death determined by LDH assay do not always give a comparable result, for example, in the case that cells do not undergo cell death but do not grow. We explained this potential discrepancy in the revised manuscript (line 189-192).

2. Please discuss why sodium chlorate (or reduction of sulfation in ECM) treatment did not increase background ROS generation (e.g. Figure 5b).

We rewrote the discussion as follows:

Ali et al. proposed that chlorate enters the erythrocyte, probably through band 3 anion channel, where it is reduced to chloride ion and this process generates ROS and reactive nitrogen species. However, at present, it is uncertain whether band 3 anion channel can function in HT22 cells (line 329-332).

3. What is the interpretation of increased phosphop54JNK but not phosphop46JNK following cotreatment with erastin and sodium chlorate over erastin alone (Fig. 6)?

Although there is no statistical significance of phospho-p54 JNK between erastin alone and erastin plus sodium chlorate, the responses of phospho-p46 JNK and phospho-p54 JNK to those treatments are similar. The signal of phospho-p46 JNK was so strong, therefore, the intensity may not be proportional to the level of phospho-46 JNK protein.

Will inhibiting JNK but not Nrf2 ameliorate erastin + sodium chlorate toxicity and cell death in HT22 cells?

We have reported previously that a pharmacological inhibitor of JNK SP600125 does not have a prominent protective effect on glutamate- and erastin-induced cell death (Hirata et al., Toxicology 418, 1-10, 2019, 2019). This is explained by the fact that binding of activated JNK to non-transcriptional targets such as the outer mitochondrial membrane protein Sab leads to increased mitochondrial ROS and mediates ER stress-induced apoptosis in cultured cells and ischemic necrosis in the brain (Chambers et al., J. Biol. Chem. 288, 1079-1087, 2013; Win et al., Cell Death Dis. 5, e989, 2014). Therefore non-transcriptional targets of phospho-JNK could play a significant role not only in ER stress but also in oxytosis and ferroptosis.

4. If diffusion of divalent cation (e.g. Fe2+) by HSPGs and CSPGs may play a critical role in the toxicity described in this study, will cotreatment with Fe chelator mitigate enhanced toxicity by sodium chlorate?

We carried out additional experiments using an iron chelator, deferoxamine, and added the data as Figure 6b.

Minor points:

1. Please label panel (b) in Figure 4.

Corrected.

2. Please define system Xc as cysteine/glutamate antiporter on first appearance.

Defined system Xc as a cystine/glutamate antiporter.

3. Line 299-304 where the authors are explaining the mechanism of action of sodium chlorate is a bit confusing. Please edit this.

We rewrote this part.

4. Line 310-313 is a repeated information found in the introduction, and citations found around (e.g. #2, 20, and 14) are using ( ) instead of [ ].

Line 310-313 in the original manuscript was removed.

() was corrected to [].

Round 2

Reviewer 1 Report

The authors have satisfied my previous criticism.

Reviewer 2 Report

The authors addressed most of concerns raised in the original submission. I do not have any further concerns.